# Computing Strategic Responses to Non-Linear Classifiers

## Abstract

We consider the problem of strategic classification, where the act of deploying a classifier leads to strategic behaviour that induces a distribution shift on subsequent observations. Current approaches to learning classifiers in strategic settings are focused primarily on the linear setting, but in many cases non-linear classifiers are more suitable. A central limitation to progress for non-linear classifiers arises from the inability to compute best responses in these settings. We present a novel method for computing the best response by optimising the Lagrangian dual of the Agents' objective. We demonstrate that our method reproduces best responses in linear settings, identifying key weaknesses in existing approaches. We present further results demonstrating our method can be straight-forwardly applied to non-linear classifier settings, where it is useful for both evaluation and training.

## 1  Introduction and Related Work

Biased training data can have a significant adverse impact on the accuracy and fairness of models when they are applied to future observations that come from a different distribution. In classifier learning, one possible source of bias arises from tactical misrepresentation by people who interact with the classifier; those with a preference for a certain classification, and a knowledge of the classifier that is being used, have an incentive to misrepresent their state to receive a favourable outcome. This dynamic can arise in many sociotechnical situations, spanning from universities deciding which students to enrol as far as institutions determining how to distribute aid and resources in times of need.

The Strategic Classification [9, 2] literature examines the setting where a *Learner* is attempting to classify a population of *Agents* in that can independently perturb their representations in response to the learned classifier. The principal objective of the Learner is to develop a classifier that is robust to the Agents' perturbations, thereby reducing their ability to undermine the overarching goal of the Learner. This is typically formulated in the language of game theory: the Learner takes the role of a leader in a Stackelberg game, and the Agents follow with their best response. Computing the best response presents a considerable challenge in Strategic Classification; its definition is typically discontinuous and non-differentiable in all but the simplest cases, which makes it unwieldy to work with. As a consequence of this impediment, most of the research that has been done in Strategic Classification has been focused on linear classifiers, where one has a closed form expression for the best response [13, 14, 5]. This limits the diversity and the scope of the research that can be performed, as well as the degree to which developments in this area can be plausibly adopted and applied in real-world settings.

One algorithm that can be used for approximating the best response to non-linear classifiers is Repeated Empirical Gradient Descent (REGD), as proposed in Perdomo et al. [17]. Arising from the related field of Performative Prediction, the authors demonstrate that it can be applied to the Strategic

Classification setting. While the method in Perdomo et al. [17] can be used for non-linear classifiers, they focus on the case of linear classifiers. Subsequent works, such as Izzo et al. [11], Mofakhami et al. [16], Zhong et al. [21], have iterated on this algorithm to work on a more generalised performative settings. However, there approaches utilise methods such as counterfactual reasoning to make inferences about the behaviour of the best response, and generally avoid explicitly computing it.

In this work we present a method for computing strategic responses to non-linear classifiers that is less conservative and more cost-efficient than the approach of Perdomo et al. [17]. Building on recent iterative methods from the field of Performative Prediction, we also demonstrate that our method results in a learning algorithm that is capable of producing strategically robust classifiers for a range of differentiable classifier families.

## 2   Strategic Classification and the Best Response

For a feature space, $\mathcal{X}$, and label space, $\mathcal{Y} = \{-1, 1\}$, let $\mathcal{D}$ be a probability distribution on their Cartesian Product, $\mathcal{X} \times \mathcal{Y}$. Given a class of models, $\mathcal{H} \subset \mathbb{R}^{\mathcal{X}}$, the usual goal when training a classifier is to find some model, $f \in \mathcal{F} = \{\boldsymbol{x} \mapsto \text{sgn}(h(\boldsymbol{x})) : h \in \mathcal{H}\}$, that achieves high accuracy. Given access to a training dataset, $S = \{(\mathbf{x}_i, \mathbf{y}_i)\}_{i=1}^n$, consisting of independent and identically distributed samples from $\mathcal{D}$, this goal is often addressed by finding a model that minimises the empirical risk,

$$\hat{f} = \arg\min_{f \in \mathcal{F}} \frac{1}{n} \sum_{i=1}^{n} l(f(\mathbf{x}_i), \mathbf{y}_i), \tag{1}$$

where $l$ would ideally be the zero–one loss, but is more typically chosen to be the hinge loss in order to facilitate tractable optimisation.

In many realistic classification scenarios $\mathcal{D}$ is not representative of the data that a trained classifier will actually be deployed on [18, 19]. Distribution shift can adversely impact the accuracy of $f$, since $f$ will not necessarily be optimal under the new data distribution. In a strategic setting, distribution shift can arise because Agents corresponding to some data point, $(\boldsymbol{x}, y)$, may be motivated to manipulate their representations, $\boldsymbol{x}$, in order to obtain a positive classification from $f$, irrespective of their true label, $y$. In the strategic classification literature this is typically modelled using an idealised best response function [9],

$$\Delta_f(\boldsymbol{x}) = \arg\max_{\boldsymbol{z} \in \mathcal{X}} f(\boldsymbol{z}) - c(\boldsymbol{x}, \boldsymbol{z}). \tag{2}$$

Crucially, it is assumed that an Agent must pay some cost, $c(\boldsymbol{x}, \boldsymbol{z})$, to manipulate their representation from $\boldsymbol{x}$ to $\boldsymbol{z}$. This cost is assumed to satisfy several natural properties: no manipulation should result in no cost, and the cost should be subadditive, $c(\boldsymbol{x}, \boldsymbol{z}) \leq c(\boldsymbol{x}, \boldsymbol{y}) + c(\boldsymbol{y}, \boldsymbol{z})$. Various extensions of this standard setup exist that allow for, e.g., different levels of information to be revealed to the Agents [7, 3] or for Agents to not necessarily favour positive classification [14].

This strategic interaction between the Agents and the Learner results in a breakdown in performance of the classifier [13]. To address this, algorithms have been designed to support training of classifiers that are robust to such misrepresentations [13, 20, 17]. In particular, the idea of adapting the empirical risk to include this idealised best response is natural. This has lead to the Strategic Empirical Risk Minimisation (SERM) approach [20],

$$\hat{f} = \arg\min_{f \in \mathcal{F}} \quad \frac{1}{n} \sum_{i=1}^{n} l(f(\Delta_f(\mathbf{x}_i)), \mathbf{y}_i) \tag{3}$$
$$\text{subject to} \quad \Delta_f(\mathbf{x}_i) = \arg\max_{\boldsymbol{z} \in \mathcal{X}} f(\boldsymbol{z}) - c(\mathbf{x}_i, \boldsymbol{z}),$$

a bilevel optimisation problem, where the $\arg\min$ over $f$ is referred to as the upper level and the $\arg\max$ over $\boldsymbol{z}$ as the lower level. For a linear model parameterised by $\boldsymbol{w}$ and the Euclidean distance cost function, this collapses to a single level problem due to the closed form best response [4, 17],

$$\Delta_f(\boldsymbol{x}) = \begin{cases} \boldsymbol{x} - \boldsymbol{w}\frac{h(\boldsymbol{x})}{\|\boldsymbol{w}\|_2} & \text{if } -2 \leq h(\boldsymbol{x}) < 0 \\ \boldsymbol{x} & \text{otherwise.} \end{cases} \tag{4}$$

In cases where a closed form is not available, one can employ the Repeated Empirical Gradient Descent (REGD) approach of Perdomo et al. [17] to obtain an approximation of the empirical risk minimiser, $\hat{f}$. This requires black box access to a routine that computes a strategic response to some classifier, $f$.

## 3 Responding to Non-Linear Classifiers

When the underlying classifier is non-linear, or the cost is less amenable to optimisation than the squared Euclidean distance, we cannot expect to find a global optimum of Equation 2. In this Section we discuss methods for computing acceptable approximations to the best response.

**Gradient Descent Response**    Gradient-based methods are a common approach for solving optimisation problems such as Equation 3. Such methods involve using gradient ascent to solve the lower level maximisation, and gradient descent to solve the upper level minimisation. Perdomo et al. [17], propose an iterative gradient-based method that they show can be used to approximate solutions to Strategic Classification problems.[1] However, we note that, since $f$ is a binary classifier, it is not differentiable, and so gradient methods cannot be directly applied to the problem as presented in Equation 3.

In the case where $\mathcal{H}$ is a class of differentiable models, one approach to allow for gradient-based solutions concepts is to replace $f$ in the lower objective with the corresponding differentiable $h$,

$$\Delta_f^{GD}(\boldsymbol{x}) = \arg\max_{\boldsymbol{z} \in \mathcal{X}} h(\boldsymbol{z}) - c(\boldsymbol{x}, \boldsymbol{z}). \tag{5}$$

This is effectively what is done in [17]. However, as a consequence of this relaxation, one of the constraints on the best response is relaxed; in the best response of Equation 2, solutions trade off maximising $f(\boldsymbol{z})$, minimising the cost $c(\boldsymbol{x}, \boldsymbol{z})$, and constraining the cost to be less than two ($c(\boldsymbol{x}, \boldsymbol{z}) \leq 2$). This last constraint arises from the interaction between the utility and the cost terms in the optimisation; $f$ is binary and the maximum the utility can be improved by is two. If the cost to realise this exceeds two then the Agent would realise a higher objective by letting $\boldsymbol{z} = \boldsymbol{x}$. This constraint does not necessarily apply in the case where $f$ is replaced with $h$.

**Lagrangian Dual Response**    In this work we consider an alternative formulation of the lower level objective. By treating it explicitly as a constrained optimisation problem, where the objective is to minimise the cost, we can write Equation 2 equivalently as,

$$\Delta_f(\mathbf{x}) = \arg\min_{\mathbf{z} \in \mathcal{X}} \quad c(\mathbf{x}, \mathbf{z}), \tag{6}$$
$$\text{subject to} \quad h(\mathbf{z}) \geq 0,$$
$$c(\mathbf{x}, \mathbf{z}) \leq 2.$$

This formulation captures the same objectives that applied in Equation 2, but it does so without depending on the interaction between the utility and the cost during the optimisation. In practice, one must constrain $h(\boldsymbol{z}) \geq \epsilon$ for some small $\epsilon$ to ensure the inequality holds strictly.

The problem formulation in Equation 6 is amenable to Karush-Kuhn-Tucker (KKT) solution methods [12, 8]. Under this formulation we can replace the objective in Equation 6 with the Lagrangian dual,

$$\mathcal{L}(\boldsymbol{x}, \boldsymbol{z}, \lambda, \mu) = c(\boldsymbol{x}, \boldsymbol{z}) - \lambda(h(\boldsymbol{z}) - \epsilon) - \mu(c(\boldsymbol{x}, \boldsymbol{z}) - 2), \tag{7}$$

where $\lambda, \mu \in \mathbb{R}$ are Lagrange multipliers. This is used to define an optimisation problem,

$$\Delta_f^{LD}(\mathbf{x}) = \arg\min_{\mathbf{z}} \max_{\lambda, \mu} \quad \mathcal{L}(\mathbf{x}, \mathbf{z}, \lambda, \mu) \tag{8}$$
$$\text{subject to} \quad \lambda, \mu \geq 0,$$

that can be solved with projected gradient ascent-descent. The projection function simply clamps the $\lambda$ and $\mu$ Lagrange multipliers to ensure they remain non-negative. The solution to the Langrangian dual optimisation converges to a locally optimal solution whilst enforcing the constraints. In the case where all local optima are global optima—e.g., if $h$ and $c$ are convex in $\boldsymbol{z}$—this approach is guaranteed to compute the best response.

**Post-Response Checks**    Valid responses must satisfy two invariants, $f(\Delta_f(\mathbf{x})) > 0$ and $c(\mathbf{x}, \mathbf{z}) < 2$. Both approaches described above may violate these invariants in some instances. The Gradient Descent response will sometimes incur more cost than is rational, given the reward gained from obtaining a positive classification. The Lagrangian Dual may violate one of these constraints if there is no feasible point. To avoid these issues, we manually check that both invariants are satisfied and return the original $\boldsymbol{x}$ if they are not.

---

[1] They prove that the convergence of their method to optimal solutions is limited to a narrow family of classifiers

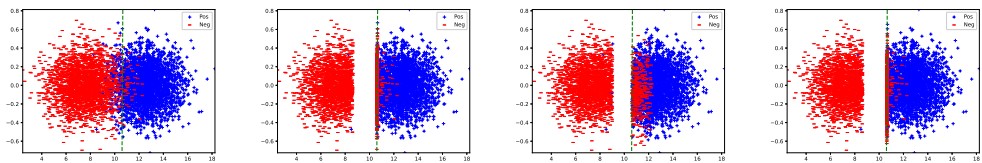

Figure 1: Various strategic responses to a fixed linear classifier applied to Gaussian dataset; Left: Linear SVM decision boundary; Middle Left: Ground Truth Response; Middle Right: Gradient Descent Response; Right: Lagrangian Dual Response (Ours)

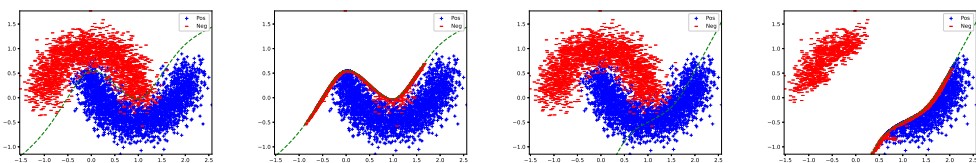

Figure 2: Left: MLP Classifier trained on twin moons with ERM; Left Middle: Lagrangian Response to ERM MLP model; Middle Right: MLP Classifier trained with REGD method; Right: Lagrangian Response to REGD MLP model.

## 4 Experimental Results

**Responding to Linear Models**    In order to qualitatively examine and compare different response computations methods we examine their impact on a simple toy dataset. We consider a two dimensional dataset comprised of two Gaussians; one corresponding to the negative class, and the other to the positive. Both classes have the same number of data points. We train a linear support vector machine (SVM) on the dataset, without taking any effort to make the classifier strategically robust. We consider three methods for computing strategic responses: the exact solution given in Equation 4, the Gradient Descent response in Equation 5, and the Lagrangian Dual response from Equation 8. For all three approaches we use the Euclidean distance as the cost function.

Figure 1 provides a visual comparison of these response methods on the Gaussians dataset. The left pane shows the unperturbed dataset along with the location of the SVM decision boundary. The Left-Middle pane demonstrates the best response behaviour arising from using the closed form expression in Equation 4. The Middle-Right and the Right panes are the Gradient Descent and Lagrangian Dual responses, respectively. Comparing with the optimal response, we observe that the Gradient Descent approach is susceptible to two types of errors: (i) there are some points that move too far over the decision boundary, meaning they incur more cost than is necessary; and (ii) some points that could pay a cost close to, but still less than, two are not manipulated. In contrast to this, our Lagrangian Dual approach is able to exactly replicate the behaviour of the true best response.

**Responding to Non-Linear Models**    We investigate qualitatively whether our Lagrangian Dual response is able to successfully respond to a classifier trained with ERM, and another trained using REGD with our approach also used as the response function. This is achieved by training a small Multi-Layer Perceptron (MLP) model on the twin moons toy dataset.

Figure 2 shows the results of this experiment. The Left pane visualises the ERM decision boundary and the unperturbed dataset. While this is optimal in absence of strategic behaviour, all of the negative points are close to the decision boundary, and our strategic response is therefore able to move them to lie on the positive side of the decision boundary, as demonstrated in the Middle Left pane. The Middle Right pane depicts the decision boundary when the model is trained with REGD. Accuracy on the unperturbed data points is poor, but when the strategic behaviour is applied in the Right pane, we see that all the positive points are classified correctly but only a subset of the negatively labelled points successfully manipulate their features.

**Quantitative Comparison**    We provide a quantitative comparison on the GiveMeSomeCredit dataset [6], a popular testbed in the Strategic Classification literature due to the incentives and capabilities of loan applicants to strategically manipulate their features [10]. The dataset details

Table 1: Percentage of correct predictions (mean ± standard error) made by linear models (top), ICNN model (middle), and MLP model (bottom) on the Give Me Some Credit Dataset when different strategic response methods are used during training (rows) and testing (columns).

|  | $\Delta_f$ | Identity | Gradient | Lagrange |
|---|---|---|---|---|
| Linear | I | $59.00 \pm 0.38$ | $52.06 \pm 0.39$ | $50.34 \pm 0.39$ |
|  | GD | $47.46 \pm 0.39$ | $49.55 \pm 0.39$ | $49.84 \pm 0.39$ |
|  | LD | $57.65 \pm 0.38$ | $57.81 \pm 0.38$ | $50.15 \pm 0.39$ |
| ICNN | I | $63.33 \pm 0.37$ | $67.62 \pm 0.36$ | $50.04 \pm 0.39$ |
|  | GD | $63.73 \pm 0.37$ | $63.78 \pm 0.37$ | $50.86 \pm 0.39$ |
|  | LD | $64.65 \pm 0.37$ | $63.97 \pm 0.37$ | $51.21 \pm 0.39$ |
| MLP | I | $73.59 \pm 0.34$ | $50.41 \pm 0.39$ | $50.00 \pm 0.39$ |
|  | GD | $69.16 \pm 0.36$ | $50.21 \pm 0.39$ | $50.08 \pm 0.39$ |
|  | LD | $62.49 \pm 0.37$ | $62.64 \pm 0.37$ | $54.33 \pm 0.39$ |

Table 2: Percent of points gamed (mean ± standard error) for linear models (top), ICNN model (middle), and MLP model (bottom) on the Give Me Some Credit Dataset for different strategic response methods.

|  | $\Delta_f$ | Gradient | Lagrange |
|---|---|---|---|
| Linear | I | $42.59 \pm 0.38$ | $48.72 \pm 0.39$ |
|  | GD | $60.74 \pm 0.38$ | $66.00 \pm 0.37$ |
|  | LG | $75.32 \pm 0.33$ | $88.84 \pm 0.24$ |
| ICNN | I | $26.60 \pm 0.34$ | $79.80 \pm 0.31$ |
|  | GD | $22.47 \pm 0.32$ | $57.91 \pm 0.38$ |
|  | LG | $16.99 \pm 0.29$ | $58.43 \pm 0.38$ |
| MLP | I | $60.76 \pm 0.38$ | $61.16 \pm 0.38$ |
|  | GD | $61.90 \pm 0.38$ | $62.16 \pm 0.38$ |
|  | LG | $44.18 \pm 0.38$ | $69.58 \pm 0.36$ |

information about people with the objective of predicting whether they will experience imminent financial distress. Three types of classifiers are trained: linear, MLP, and Input Convex Neural Networks [1]. ICNNs are MLPs that are constrained to ensure they implement functions that are convex with respect to the input features. We train the models to be strategically robust by minimising the cross entropy using the REGD training algorithm. We instantiate REGD with three different response functions: the identity, $\Delta_f^I(\boldsymbol{x}) = \boldsymbol{x}$; the Gradient Descent response $\Delta_f^{GD}$; and the Lagrangian Dual response $\Delta_f^{LD}$. Each model is evaluated by examining how robust they are to the each of the response definitions.

Table 1 contains the results of our experiments. Each row of the table corresponds to a different model instance, and each column refers to the response definition used in the evaluation. We find that robustness promoting training can adversely impact on the model performance on the unperturbed dataset. However, models trained to be robust to the Lagrangian Dual response are consistently more strategically robust than models trained under either Identity or Gradient Descent responses. This result is support by Table 2, which shows the proportion of the dataset that successfully responds the classifier in each experiment. We see that, for all model types, the Lagrangian Dual response consistently identifies more points that can respond the classifier than the Gradient Descent approach. The only points that can be perturbed by either response method are points which would be perturbed under the "true" best response, $\Delta_f$. This is a consequence of the post-response checks described in Section 3. This indicates that the Lagrangian Dual response approximates the best response with greater recall than the Gradient Descent response.

## 5   Discussion & Conclusion

Strategic behaviour is a very accessible means of inducing bias in data distributions, which can in turn undermine the accuracy and fairness of classifiers applied to this data. If it is not appropriately addressed it can facilitate malicious actors in compromising the ability of institutions to appropriately allocate resources. Therefore it is important that methods exist for producing classifiers that are robust to these behaviours. In this work we have presented a novel method for approximating strategic responses to non-linear classifiers, and have demonstrated that it can be effectively utilised to produce strategically robust non-linear classifiers.

Our intention in this work is to highlight the benefits that can be realised by producing models that are robust to strategic behaviour. However, as is apparent from our results (Figure 2 and Table 1), producing models that are robust to strategic behaviour can have the inadvertent consequence of negatively misclassifying some people. These people, who otherwise may not have been inclined to behave strategically, thereby have no option but to attempt to manipulate the classifier, or else risk misclassification. This can put undue financial (or otherwise) burden on people. While some work has been done to explore this consequence in strategic settings (e.g., [15]), an adequate resolution has yet to be found.

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
