# OpenReview forum: "Computing Strategic Responses to Non-Linear Classifiers"
_EurIPS.cc/2025/Workshop/UPLB — UPLB2025_

### Official Review · Reviewer_aBKs · 2025-10-25
**Computing Strategic Responses to Non-Linear Classifiers**

**Rating:** 7
**Confidence:** 3

**Review:**

The authors study the problem of how to improve the performances of a binary classifier when the training data is corrupted. This problem has been already considered in the literature and it is relevant for the scope of the workshop. The main contribution of the paper concerns the algorithmic strategy to solve the training problem. I think that the paper contains an interesting contribution and is well written.

---

### Decision · Program_Chairs · 2025-11-03

Accept (Poster)